# Analysis of Bio-Risk Management System Implementation in Indonesian Higher Education Laboratory

**DOI:** 10.3390/ijerph18105076

**Published:** 2021-05-11

**Authors:** Anom Bowolaksono, Fatma Lestari, Saraswati Andani Satyawardhani, Abdul Kadir, Cynthia Febrina Maharani, Debby Paramitasari

**Affiliations:** 1Cellular and Molecular Mechanisms in Biological System (CEMBIOS) Research Group, Department of Biology, Faculty of Mathematics and Natural Sciences, Universitas Indonesia, Depok 16424, Indonesia; 2Disaster Risk Reduction Centre (DRRC), Universitas Indonesia, Depok 16424, Indonesia; fatma@ui.ac.id (F.L.); saraswatiandanis@gmail.com (S.A.S.); abdul_kadir@ui.ac.id (A.K.); debbyparamitasari@gmail.com (D.P.); 3Occupational Health and Safety Department, Faculty of Public Health, Universitas Indonesia, Depok 16424, Indonesia; 4Occupational and Environmental Health Department, Public Health Faculty, University of Iowa, Iowa City, IA 52242, USA; cynthia-maharani@uiowa.edu

**Keywords:** bio-risk management, higher education, laboratory, gap analysis, ISO 35001:2019

## Abstract

Developing countries face various challenges in implementing bio-risk management systems in the laboratory. In addition, educational settings are considered as workplaces with biohazard risks. Every activity in a laboratory facility carries many potential hazards that can impact human health and the environment and may cause laboratory incidents, including Laboratory Acquired Infections (LAIs). In an effort to minimize the impact and occurrence of these incidents, it is necessary to evaluate the implementation of a bio-risk management system in every activity that involves handling biological agents. This study was conducted in an Indonesian higher-education institution, herein coded as University Y. This is a descriptive, semi-quantitative study aimed at analysing and evaluating the implementation of the bio-risk management systems used in laboratories by analysing the achievements obtained by each laboratory. The study used primary data that were collected using a checklist which referred to ISO 35001:2019 on Laboratory Bio-risk Management. The checklist consisted of 202 items forming seven main elements. In addition, secondary data obtained from literature and document review were also used. The results show that out of 11 laboratories examined, only 2 laboratories met 50% of the requirements, which were Laboratory A and B, achieving good performance. Regarding the clauses of standards, a gap analysis identified leadership, performance evaluation, and support as elements with the lowest achievement. Therefore, corrective action should be developed by enhancing the commitment from management as well as improving documentation, policy, education and training.

## 1. Introduction

Laboratories in a higher education are usually used to conduct and support research and other academic activities. However, any laboratory, including ones in higher education, carries various risks and hazards. At the same time, they are also prone to emergency situations such as fire, chemical spills, infectious or biological agent contamination, and toxins [1]. Further, biological materials are considered as a main cause of incidents occurring in this field, particularly Laboratory Acquired Infections (LAIs), which can contaminate the surrounding environment. The evidence of laboratories’ incidents has been well documented. For example, the West Nile virus with SARS contamination infected a postgraduate student in Singapore in 2003 [2]. Other cases include the Ebola outbreak in Russia in 2004, Severe Acute Respiratory Syndrome (SARS) outbreak in Beijing, and tularaemia in Boston in 2004 [3].

Laboratories are the centre of public health investigations, which aim to provide services to clients and patients in implementing certain public health programs using evidence-based decisions. Various institutions such as medical hospitals and academic institutions, both public and private, contribute to diagnostic activities in different scope of works [4]. In addition to the capability to perform diagnostic activities, laboratories also need to have sufficient capabilities to conduct their work in a safe and secure manner. Since the impact of biological agent contamination can be enormous, hospitals, educational institutions and companies have tried to enhance their efforts in preventing damages to humans and facilities, both accidentally or intentionally [5]. In fact, the Occupational Safety and Health Administration (OSHA) reported that higher education laboratories are 11 times more treacherous as opposed to laboratories in industrial facilities [1]. Hence, a bio-risk management system should be applied in order to minimize all perceived risks, including risks found in educational settings. Given the current global situation, the occurrence of Coronavirus Disease 2019 (COVID-19), which was initially discovered in Wuhan, has alarmed public health authorities around the world including in Africa, South East Asia, America, Europe, and the Western Pacific, and has posed a new threat to humans. In fact, the virus has significantly escalated over time and was declared a global pandemic by the World Health Organization (WHO) on 11 March 2020 [6]. The rapid spread of this virus requires laboratories to have the capability to provide immediate diagnostic testing for COVID-19 suspects, which is considered an integral part of disease control [7]. In addition, this recent pathogen emergence forced laboratories to strengthen and review their bio-risk practices and update them to include novel recommendations [8]. Moreover, in order to conduct research safely during this pandemic, it is necessary to implement a bio-risk management system (BRM) [9,10] which encompasses two crucial elements, which are biosafety and biosecurity. The former tends to focus on procedures and techniques in the laboratory, such as preventing accidents, whereas the latter focuses on accountability measurements and procedures to protect biological substances from unauthorized access [11,12]. According to the 4th Laboratory Biosafety Manual proposed by the World Health Organization, it is crucial that biosafety and biosecurity activities are managed properly in order to protect laboratory employees and surrounding communities against unintentional exposures to, or the spread of, pathogenic biological agents. Hence, a risk assessment framework needs to be implemented as a control measure to minimize the possibility and consequences of any potential exposure to biological agents in laboratories. This framework started from gathering information to review risks and risk control measures [13].

BRM emphasizes the roles and responsibilities of each interested party working in the laboratory and ensures that the highest level of management has the most fundamental roles and responsibilities in the laboratory. In addition, the bio-risk management system is also made a priority. This system includes an evaluation of all risks, including all operations in laboratory facilities. The bio-risk management system consists of bio-risk identification, bio-risk assessment, bio-risk management, and bio-risk communication [14]. According to Wu [15], several strategies on laboratory biosafety can be applied by establishing a primary network of high-level biosafety laboratories, utilization of biological resources, training and development of professional teams, document control, high level prevention and control efforts against diseases, and international collaboration. Furthermore, regulation is also one of the main strategic visions for an effective and comprehensive biological risk management system [16].

However, there are several challenges in the BRM implementation in laboratories. Firstly, biological safety aspects in the laboratory have received little attention. For example, handling biological materials, as well as handling specimen waste and biological cultures, has not been much discussed. These issues need further analysis to determine the level of hazards, both for laboratory workers and for the surrounding environment [17]. Secondly, laboratory infections usually occur due to a lack of awareness and untrained personnel, both of which could trigger at-risk behaviours while working in the laboratory [18]. Thirdly, a lack of capacity in implementing biosafety practices was observed, mostly in developing countries [19]. It has been reported that the Southeast Asia region is the most risky area for emerging diseases, particularly in overpopulated and low-income countries [20]. Lastly, unavailability or lack of design and practice in handling infectious agents in relevant facilities has been identified in developing countries [21].

A current standard has been set up on bio-risk management, namely the International Organization for Standardization (ISO) 35001:2019, which provides a comprehensive element on bio-risk management for laboratories and other related organisations. This standard encompasses several elements involving the context of organisation, leadership, support, planning, operation, performance evaluation and improvement [22]. This standard also emphasizes the plan-do-check-act (PDCA) system principle, which highlights the sequence process of identification, assessment, controlling and monitoring the available risks and hazards associated with biological substances in laboratories within ten clauses, from scope to continuous improvement. Interestingly, a study conducted by Lestari et al., [23] applied this standard as the checklist instrument for assessing the implementation of bio-risk management in clinical and medical laboratories in Indonesia. However, studies that utilise this international standard are still limited, particularly in educational settings. Therefore, this study aims to conduct an initial survey on what bio-risk assessments, based on the standard requirements in ISO 35001:2019, have been implemented in an Indonesian higher education institution, which is referred to as University Y in this study. University Y has more than 214 laboratories across 13 faculties with various activities which pose potential hazards and high risks related to their use of biological agents. Thus, this risk basement provides a future opportunity to obtain a nature or best practice of bio-risk management implementation and identify the characteristics of risks in an educational institution. In due course, prevention and control measures can be proposed as recommendations.

## 2. Materials and Methods

### 2.1. Study Design

This study is a semi-quantitative descriptive study that analyses the bio-risk implementation evaluation in laboratories of University Y. Primary data were collected through direct observation, a BRM instrument checklist, and an interview of the head of laboratories. The survey was performed by a principal investigator and a professor whom is an expert in Occupational and Health Safety and certified in BRM. Moreover, the secondary data were obtained through literature review. The study was performed during the period of May–June 2020.

### 2.2. Study Subjects

The present study was carried out at an Indonesian higher education institution, or University Y. This institution is one of the biggest universities in the country, and was established in 1849. The university currently has 13 faculties which are categorized into several clusters: the health science cluster, science and technology cluster, and social humanities. One of the supporting facilities for these faculties is laboratories. Hence, the unit of analysis was laboratories managed by faculties, with a total number of 214 laboratories. It has been observed that University Y has two types of laboratories. The first type is the wet laboratory, where activities using chemicals and biological agents are held. The university has 90 laboratories in this category. The second type is the dry laboratory that does not use chemicals and biological agents in its activities. An example of a dry laboratory is the computer laboratory. The inclusion criterion used in this study was the laboratories that involved the use of biological materials in their activities, resulting in 11 laboratories being included as the subjects in this study. These laboratories are Lab A (Bioprocess) located in Bioprocess Engineering, Department of Chemical Engineering, Faculty of Engineering; Lab B (Microbiology) from Environmental Engineering, Department of Civil Engineering, and Faculty of Engineering; Lab C (Micro Application), Lab D (Plant Tissue Isolation), Lab E (Preparation), Lab F (Instrument), Lab G (Animal Taxonomy), and Lab H (Marin Bio) which are located in the Department of Biology Faculty of Mathematics and Natural Sciences; and Lab I, J, K (Wet Lab 1, 3, 4) in the Health Science Cluster. Since these final three laboratories are under the same management and share characteristics, they were grouped into one category.

### 2.3. Instrument and Data Analysis

The data collection was carried out using a laboratory assessment checklist which was based on ISO 35001:2019. It highlighted the plan-do-check-act (PDCA) system principle. This ISO standard points out the process used to identify, assess, control and monitor the risks and hazards associated with biological substances in ten clauses, from scope to improvement. The total of 202 checklist items derived from seven main clauses in ISO 35001:2019 were observed [23] (Table 1). The gathering of data was based on the result of BRM instrument checklist that had been assessed regarding the implementation of each clause. Then, data processing was done by calculating each of the yes (meeting all requirements with documented evidence of full implementation), partial (not meeting all requirement fully), and no (No evidence of implementation) responses with a score of 1, 0.5 and 0 respectively. The validation of implementation was observed through the evidence of documents, documented programs and so on during the survey. The total score was calculated using the following formula:$$\mathrm{Total}\;\mathrm{implementation}\;\mathrm{score}\;\left(T\right)=\frac{\mathrm{Score}\;\mathrm{obtained}\;}{\mathrm{Total}\;\mathrm{possible}\;\mathrm{scores}}\times 100\%$$

The maximum total score that could be obtained was 100%. The calculation process was performed using Microsoft Excel. After data processing was performed for all laboratories, a gap analysis was performed. The gap analysis aimed to identify how many and what aspects were not implemented among the 202 items. Analysis in this study used the univariate analysis with descriptive study and each achievement met by the laboratory was presented by element in a chart. According to Naroeni et al., an achievement total is categorised to be good if the laboratory reaches a total score of 50% [11].

### 2.4. Ethical Considerations

This study was approved by the ethics committee of the Research and Community Engagement of Faculty of Public Health, Universitas Indonesia under the Ethical Clearance No: Ket-99/UN2.F10.D.11/PPM.00.02/2020.

## 3. Results

Observation was undertaken on the implementation of the bio-risk management systems in 11 laboratories in University Y. The fulfilment was calculated as a percentage by involving seven clauses, namely organizational context, leadership, planning, support, operations, performance evaluation and improvement. Overall, of the 11 laboratories examined, only 2 of them had a score higher than 50%. Lab A and B obtained the highest scores, which were cited at 55% and 58%, respectively, while other laboratories had scores ranging from 33% to 44% (Figure 1). More details for each clause are described in Table 1. According to surveys, sub-clauses where no evidence of implementation was identified were highlighted in red.

In clause 4, the context and organization encompassed seven items, which were divided into four items that involved understanding the organization and its context, understanding the needs and expectations of interested parties, determining the scope of bio-risk management systems, and the bio-risk management system. Of the 11 laboratories, only 2 laboratories conformed to 6 of 7 items, namely Laboratory A and B (71%), whereas other laboratories only conformed to 1 of the 7 items (14%). It was also observed that most laboratories had not set up or determined any bio-risk management systems.

In the leadership clause (clause 5), there were still many laboratories that could not meet the requirements properly. Of the 50 items, only about 16–28% were met. The clause was divided into three parts: leadership and commitment, policies and roles, responsibilities, and authority. The majority of laboratories had not established related policies.

As far as clause 6 was concerned, the planning clause consisted of 11 items. These items were divided into two: actions to address risks and opportunities, and bio-risk management goals and plans to achieve them. In the first part, questions were asked related to risk management in the laboratory, and the second part asked about quality objectives and information related to bio-risk. The results show that Laboratories D and G had not implemented this clause properly, with a percentage of conformity of 9% and 0%, respectively. In more details, each of these laboratories only fulfilled 1 and 0 item (out of 11 items), respectively. In contrast, Laboratory B, H, I, J and K achieved a higher score of 55%.

In clause 7, the support clause consisted of 55 items which were categorized into 8 sub-clauses: resources, competence, awareness, communication, documentation, non-employees, personal security and supplier control. The majority of laboratories obtained a score of more than 50%, except for laboratories G and H. It was clearly seen that a lack of documented information was identified in all laboratories. In fact, for the personal competency sub-clause, only four laboratories had provided training for laboratory technicians, i.e., laboratories in medical science or Lab I, J, K. Hence, this contributed to highest score at sub-clause 7.2 (93%).

The operation clause was comprised of 43 items, which were divided into 10 sub-clauses of operational planning and control; commissioning and decommissioning; maintenance, control, calibration, certification and validation; physical security; inventory of biological materials; good microbiological technique; decontamination and waste management; emergency response and emergency planning; and transportation of biological materials. The achievements of all laboratories in this clause were conscientiously good. This was apparent from the achievement scores which were already above 50.

Generally, the performance evaluation clause describes the laboratory assessment process, which includes monitoring, measurement, analysis and evaluation; internal audit; and management review. The results of achievement for this clause were poor, because almost all laboratories had not carried out a specific internal audit for their bio-risk management system. The total score of this element ranged from 9–15%.

In clause 10, the improvement element consisted of three main sub-clauses: general; incidents, non-conformities and corrective action; and continuous improvement. The achievements of this clause varied, with 5 out of 11 laboratories succeeding in achieving a score above 50%, while the remaining 6 laboratories obtained a score below 50%.

On the results for each clause above, a gap analysis was carried out in order to identify the aspects that were not met yet so that they could be prioritized immediately for continual improvement. Gap analysis identified the items that had been not implemented from 202 items of ISO 35001:2019. As far as the gap analysis score was concerned, the three clauses with the highest gaps were leadership, performance evaluation, and support (Table 2). Furthermore, the highest gaps of leadership were identified in policies and organizational roles, responsibilities, and authorities. Meanwhile, the lowest gaps were in the clauses of the organizational context, planning, operations, and improvements.

## 4. Discussion

It is clear that laboratories in educational settings have the potential to negatively impact users, or even communities, due to potential incidents caused by their use of biological substances [9,24,25,26,27]. The present study aims to examine the implementation of bio-risk management systems in accordance with the recent international standard of ISO 35001:2019 that observed seven main clauses: organizational context, leadership, planning, support, operations, performance evaluation and improvement.

According to the survey, overall, each laboratory needs to enhance its implementation of BRM, particularly in the aspects of leadership, support and performance evaluation. As far as each clause was concerned, in the implementation of the context and organization clause, no laboratory observed under this study has implemented a bio-risk management system. In fact, their systems have not yet been set up. Even though the activities related to bio-risk are observed, they are not in accordance with existing standards. In other words, steps have been conducted without written Standard Operational Procedures (SOPs) and there were no documents related to the BRM in the subject study. In fact, the SOPs will be a guideline to achieve the consistency of test performance and reliable outcomes in managing the bio-risk hazards, and can also induce positive behaviour when working on tasks in the laboratory and ensure that employees are able to work safely [28]. Importantly, in order to establish a laboratory bio-risk management system, the organization must be able to develop and improve that system by setting up goals and objectives. A key success element of management systems is the implementation of the planning, doing, checking, and reviewing (act) cycle. An establishment of goals and objectives is the initial step of this system, and will involve the identification of risks and hazards associated with bio-risk [29].

In the leadership clause, policy is the major contributor in effective implementation of BRM. Our finding shows that policies are still being made in general, namely under the policies related to the Occupational Health and Safety of the laboratories. This means that no specific policy is developed for bio-risk. In policy development, the first step that can be taken is to identify the existing hazards and risks in the laboratory, as the content of the policy must be in accordance with the nature and scale of risks associated with the facilities and activities in the laboratory. Actions taken by the top management demonstrate a commitment to policies on laboratory bio-risk management. These policies include protection of all personnel; risk reduction of biological agents; required regulations, information and communication; and continual improvement [30]. Moreover, those policies should be compiled in a single document. When the policy development has been completed, the policy document should be signed by top management and other relevant parties, if required. A strong management commitment can have a positive impact on biosafety and biosecurity aspects in the laboratory. Thus, the existing gaps can be managed properly [31]. It is important that the roles, responsibilities and authorities of all laboratories are clearly defined and implemented; thus, there is a need to assign committees and advisors for bio-risk management and scientific management. The appointment must be based on experience, educational background and abilities to deal with biological hazards in the laboratory, which are closely related to the implementation of a bio-risk management system. In its implementation, several approaches are needed, such as dialogue, partnership, and consultation with related parties [32] including international organizations, regional organizations, national organizations, institutions, and, at the lowest level, laboratory personnel. The parties that play the most crucial roles in the implementation are researchers and students who are often most actively involved in activities in the laboratory [33]. Thus, they have to have the abilities to implement the system. The initial approach can be made by the top management at the faculty level, which is the dean, through dialogue with the heads of the laboratory and/or the head researchers. The results of the dialogue must be communicated to the biosafety officer, laboratory assistant, researchers, and students. Hence, the implementation of a bio-risk management system can be carried out comprehensively and continuously.

In the planning clause, a gap was found in the sub-clause of bio-risk management objective. In this issue, the organization had not established a document of bio-risk management objectives at relevant functions and levels. According to the ISO 35001:2019 standard, the objective should meet the requirements such as the consistency of objectives to the implementation, how the document is reviewed, communicated, and updated. Hence, laboratories are also required to prepare risk assessment documents for each activity carried out as planning systems. In the present study, several laboratories have made the compilation, which includes laboratories that have participated in the risk assessment preparation training organized by the Occupational Safety, Health and Environment Unit. However, there are still laboratories that did not perform well because of a lack of knowledge on document making. The contributing factors for encouraging an organisation to prepare a risk management system are the use of infectious materials, toxins and other hazardous materials; the existence of SOPs for practical and research implementation; new laboratory equipment and personnel; Personal Protective Equipment (PPE) providers; waste management; work accident, outbreaks and pandemics; and changes in infrastructure such as electricity and waterways. It can be said that a risk assessment plays an integral part in BRM, as the results can be used for budget planning for the following year, for example in planning a laboratory design renovation including adjusting the facilities and equipment needed. In addition, risk assessment documents can also be used to evaluate and validate emergency response plans [34]. In the aspect of goal and objective setting, the laboratory should have set these aspects in advance. Top management or the dean must conduct regular monitoring to ensure the achievement of these goals. As the first step to starting the implementation of BRM, approaches discussed earlier can be taken. In addition, dissemination of relevant materials is also recommended to be delivered to the top management and laboratories personnel to increase awareness of the risks and hazards that exist in the laboratory and the materials used. The potential supporting activities in bio-risk management can be carried out by specific programmes such as raising awareness in the context of promoting biosafety and bioethics [35].

According to clause 7, or support, practically all laboratories in this study have implemented occupational health programmes in the form of annual routine medical check-ups as one of the supporting elements in ISO 35001:2019. The programme is performed by checking the health status of the employees in the laboratory. It has been argued that the medical check-up is sufficient to analyse the health of laboratory personnel. No vaccination programme, however, has been implemented in these laboratories. In fact, the importance of vaccination for laboratory personnel is highly overriding as laboratory personnel are a vulnerable group due to exposures to biological agents on an almost-daily basis. Hence, the organization shall ensure that a vaccination policy is clearly defined and implemented [36]. The recommended vaccines are Hepatitis B, Influenza, MMR (measles, mumps, and rubella), Varicella for chickenpox, DPT (Diphtheria, Pertussis, and Tetanus) and meningococcal for meningitis. Especially for the meningitis vaccine, these should be prioritized for personnel exposed to nitrogen [37].

Another crucial aspect in the support elements is training. The competency of personnel should be considered, starting from the process of recruitment by observing the educational background, experience and certifications owned by the candidates. When the personnel are recruited and start to perform their job, they should be under regular supervision of the head of the laboratory and/or the head researcher until their competence has been proven. If the personnel cannot show the required competence, it should be contemplated whether he/she should continue working in that position. Furthermore, all laboratories are required to provide and maintain facilities, equipment and competence in handling biological substances, particularly the knowledge of biosafety level and the basic characteristics of dangerous agents [38,39]. The complexity of laboratory procedures is influenced by various factors, including the level of educational level and experience [40]; hence, the information received and knowledge of all laboratory practices and procedures should be strengthened by training programmes [41].

In the operation clause, this current study observes for the gap that the aspects of emergency response plan still need to be improved since all laboratories do not have any emergency response programmes specifically for bio-risk aspects. So far, an emergency response programme has only been established for general emergencies such as natural disasters, fires and chemical spills. There was also an absence of training regarding bio-risk emergency response, making specific training programs for this issue required. The main objective of obtaining the training is to increase the awareness of the laboratory personnel on bio-risk issues, including the relevance of human factors in bio-risk management [37]. In its implementation, emergency response is quite difficult because it requires support from several parties such as the faculty security unit, hospitals and/or health facilities at the university level, police force and others [11]. It has been clearly seen that commitment from top management is crucial, because the overall responsibility for the bio-risk management system rests with top management [12].

The performance evaluation clause represents the laboratory assessment process from monitoring to management review. However, the gap of this study observes that almost all laboratories have not carried out a specific internal audit for their bio-risk management system. Routine inspections or audits are very important in laboratory management, especially the inspection of facilities and equipment that have to be routinely maintained and checked to make sure that the equipment is standardised. Thus, risks and hazards can be controlled to the lowest level [42]. In order to encourage laboratories to carry out routine inspections and audits, it is necessary to disseminate the importance of audits. The dissemination should be addressed to the dean, as the top management at the faculty who holds the authority to plan the budget for audits. In this dissemination, it is important to convey that routine inspections or audits can provide significant improvements in terms of design, construction, operation processes and management systems [43]. The laboratories have the need for routine audits for the potential hazardous of biological materials observed [40]. In addition, top management will be able to identify deficiencies in the laboratory, which can be used as the basis for gradual improvement.

Lastly, the improvement clause delineates how the laboratory carries out the improvements based on the assessments that have been conducted. In addition, the institution needs to set up performance objectives and work on continual improvement of bio-risk management implementation [44]. Our findings show that this clause score is low due to several factors, particularly as there is no continuous improvement as a result of not carrying out internal or external audits. Hence, the laboratory has not successfully identified which aspects need to be improved. If the laboratory had carried out an internal audit, the laboratory would have the necessary records in the form of monitoring results, measurements, and audit results.

To summarize, this study proposes an overview on the nature of bio-risk management systems in an educational setting in Indonesia, with various challenges faced in implementation. However, all information obtained would have a positive impact upon other institutions developing and enhancing their awareness of BRM. On the other hand, this study comes with several limitations. Firstly, the ISO 35001:2019 does not provide a guideline for the total score of accomplishment that the score is based on references reviewed. Secondly, bias can influence the results, since the survey of this study is done through self-assessment. Thirdly, these results need to be explored further because each clause has different number of items. It is suggested that further research should be done through in-depth interviews and statistical analysis to assess the association between elements in the BRM.

## 5. Conclusions

This study underlines the current level of bio-risk management systems in an educational setting in the form of a case study in University Y. The study reported that the majority of laboratories obtain low scores, indicating the need to improve their bio-risk management systems. The three aspects with the highest gaps are leadership, support, and performance evaluation, which need to be followed up in order for the bio-risk management system in a laboratory to be fulfilled. This requires a strong commitment from top management, which is then passed on to all stakeholders involved. The improvement of these aspects will aim to protect the laboratory personnel and the surrounding environment from exposure to biological materials, and there is an urgent need for these aspects because the more a laboratory works with biological materials, the greater the risk of infection. The requirement of routine training targeted to laboratory workers, particularly on how to develop policy, bio-risk management documents, and emergency response is also identified. Importantly, a performance evaluation that entails robust action by conducting an audit and continual improvement in order to achieve the highest level of bio-risk management is needed.

## Figures and Tables

**Figure 1 ijerph-18-05076-f001:**
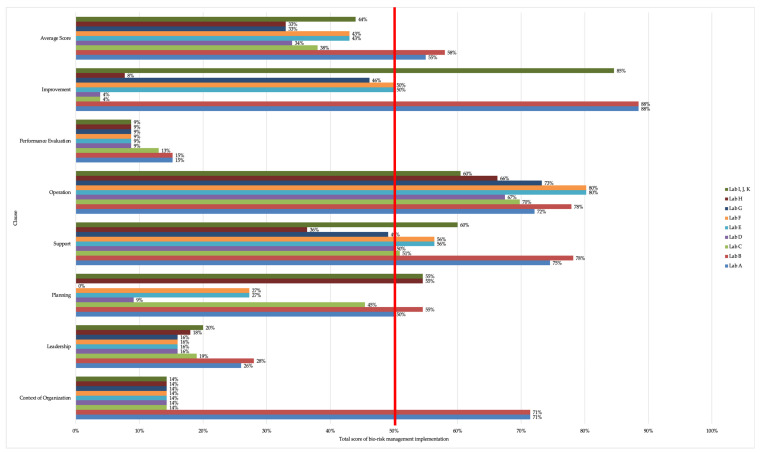
Achievements of bio-risk management system implementation.

**Table 1 ijerph-18-05076-t001:** Total implementation score (T) for each sub-clause.

No	Element	Description	Sub-Clauses		Number of Items	Lab	Lab	Lab	Lab	Lab	Lab	Lab	Lab	Lab
						A	B	C	D	E	F	G	H	I, J, K
						T	T	T	T	T	T	T	T	T
**1**	Context and Organisations	The goals and mandates of the organisation, its objectives, and the boundaries of its work must be clearly defined and communicated throughout the organisation. The organisation must determine external and internal issues that are relevant to its objectives and which affect the ability to achieve the expected results of bio-risk management system.	4.2	Understanding the Organisation and its context	2	2	2	1	1	1	1	1	1	1
			4.3	Understanding the needs and expectations of interested parties	2	1	1	0	0	0	0	0	0	0
	(Clause 4)		4.4	Determining the scope of bio-risk management systems	2	1	1	0	0	0	0	0	0	0
			4.5	Bio-risk management system	1	1	1	0	0	0	0	0	0	0
				Total	7	5 (71%)	5 (71%)	1 (14%)	1 (14%)	1 (14%)	1 (14%)	1 (14%)	1 (14%)	1 (14%)
**2**	Leadership	Top management must demonstrate leadership and commitment with respect to the bio-risk management system	5.1	Leadership and commitment	10	8.5	8.5	5.5	5	5	5	5	5	7
	(Clause 5)		5.2	Policy	8	2.5	2.5	0	0	0	0	0	0	0
			5.3	Roles, responsibilities, and authorities	32	2	3	4	3	3	3 (9)	3 (9)	4	3
				Total	50	13 (26%)	14 (28%)	9.5 (19%)	8 (16%)	8 (16%)	8 (16%)	8 (16%)	9 (18%)	10 (20%)
**3**	Planning	Actions are needed to identify, assess, and prioritize bio-risk, implement measures to mitigate bio-risk, integrate these actions into the organisation’s bio-risk management system processes, and evaluate the effectiveness of these measures.	6.1	Actions to address risks and opportunities	6	5.5	6	5	1	3	3	0	6	6
	(Clause 6)		6.2	Bio-risk management objectives and planning to achieve them	5	0	0	0	0	0	0	0	0	0
				Total	11	5.5 (50%)	6 (55%)	5 (45%)	1 (9%)	3 (27%)	3 (27%)	0 (0%)	6 (55%)	6 (55%)
**4**	Support	The organisation must carry out the necessary performance evaluations to enhance the performance of the bio-risk management system	7.1	Resources	8	1.5	3	2	2	2	2	2	1	3
			7.2	Competence	14	12	12	11	11	13	13	9	3	13
			7.3	Awareness	7	4.5	5	3	2	4	4	3	3	4
			7.4	Communication	6	6	6	6	5.5	5	5	6	6	6
	(Clause 7)		7.5	Documented information	14	12	12	1	1	2	2	2	2	2
			7.6	Non-employees	1	1	1	1	1	1	1	1	1	1
			7.7	Personal security	2	1	1	1	1	1	1	1	1	1
			7.8	Control of suppliers	3	3	3	3	3	3	3	3	3	3
				Total	55	41 (75%)	43 (78%)	28 (51%)	27.5 (50%)	31 (56%)	31 (56%)	27 (49%)	20 (36%)	33 (60%)
**5**	Operation	The organisation shall carry out the operations required for the implementation of the bio-risk management system.	8.1	Operational planning and control	10	4	4.5	4	3	7	7.5	3.5	3.5	4
			8.2	Commissioning dan decommissioning	2	2	2	2	2	2	2	2	2	2
			8.3	Maintenance, control, calibration, certification, and validation	1	1	1	1	1	1	1	1	1	1
			8.4	Physical security	4	3	3	3	3	4	4	3	4	4
			8.5	Biological materials inventory	3	3	3	3	3	3	3	3	0	0
			8.6	Good microbiological technique	2	2	2	2	2	2	2	2	2	2
	(Clause 8)		8.7	Clothing and personal protective equipment (PPE)	4	3	3	3	3	3	3	3	3	3
			8.8	Decontamination and waste management	6	4	6	3	3	3	3	4	4	4
			8.9	Emergency response and contingency planning	8	7	7	7	7	7	7	7	7	4
			8.10	Transport biological materials	3	2	2	2	2	2	2	2	2	2
				Total	43	31 (72%)	33.5 (78%)	30 (70%)	29 (67%)	34.5 (80%)	34.5 (80%)	31.5 (73%)	28.5 (66%)	26 (60%)
**6**	Performance Evaluation	The organisation must carry out the necessary performance evaluations to enhance the performance of the bio-risk management system	9.1	Monitoring, measurement, analysis, and evaluation	1	0.5 (50)	0.5 (50)	0.5 (50)	0.5 (50)	0.5 (50)	0.5 (50)	0 (0)	0 (0)	0 (0)
	(Clause 9)		9.2	Internal audit	11	11 (100)	11 (100)	0 (0)	0 (0)	6 (55)	6 (55)	6 (55)	1 (9)	11 (100)
			9.3	Management review	1	0 (0)	0 (0)	0 (0)	0 (0)	0 (0)	0 (0)	0 (0)	0 (0)	0 (0)
				Total	13	11.5 (88%)	11.5 (88%)	0.5 (4%)	0.5 (4%)	6.5 (50%)	6.5 (50%)	6 (46%)	1 (8%)	11 (85%)
**7**	Improvement	Organisations must determine opportunities for improvement from performance evaluation and implement the necessary actions to achieve the desired results of the bio-risk management system	10.1	General	1	0.5	0.5	0.5	0.5	0.5	0.5	0	0	0
	(Clause 10)		10.2	Incident, nonconformity, and corrective action	11	11	11 (100)	0	0	6	6	6	1	11
			10.3	Continual improvement	1	0	0	0	0	0	0	0	0	0
				Total	13	11.5 (88%)	11.5 (88%)	0.5 (4%)	0.5 (4%)	6.5 (50%)	6.5 (50%)	6 (46%)	1 (8%)	11 (85%)

Note: zero scores were the sub-clauses with no evidence of implementation in each laboratory. Lab A (Bioprocess) located in Bioprocess Engineering, Department of Chemical Engineering, Faculty of Engineering; Lab B (Microbiology) from Environmental Engineering, Department of Civil Engineering, and Faculty of Engineering; Lab C (Micro Application), Lab D (Plant Tissue Isolation), Lab E (Preparation), Lab F (Instrument), Lab G (Animal Taxonomy), and Lab H (Marin Bio) which are located in the Department of Biology Faculty of Mathematics and Natural Sciences; and Lab I, J, K (Wet Lab 1, 3, 4) in the Health Science Cluster.

**Table 2 ijerph-18-05076-t002:** Gap analysis of each laboratory.

Element	Lab A	Lab B	Lab C	Lab D	Lab E	Lab F	Lab G	Lab H	Lab I, J, K
**Context of organization**	2	2	6	6	6	6	6	6	6
2. **Leadership**	37	36	41.5	42	42	42	42	41	40
3. **Planning**	5.5	5	6	10	8	8	11	5	5
4. **Support**	14	12	27	27.5	24	24	28	35	22
5. **Operation**	12	9.5	13	14	8.5	8.5	11.5	14.5	17
6. **Performance evaluation**	18	18	20	21	21	21	21	21	21
7. **Improvement**	1.5	1.6	12.5	12.5	6.5	6.5	7	12	2
8. **Total**	90 (45%)	84 (42%)	*126 (62%)*	*133 (66%)*	*116 (57%)*	*116 (57%)*	*127 (63%)*	*135 (67%)*	*113 (56%)*

Note. Grey areas represent the three highest gaps found in each laboratory according to each element of ISO 35001:2019; the italics represent the labs with bad implementation of ISO 35001:2019.

## Data Availability

Data presented in this study are available upon request from the corresponding author.

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
