# Peer review of "Analysis of Bio-Risk Management System Implementation in Indonesian Higher Education Laboratory"

_ijerph, 2021, doi:10.3390/ijerph18105076_

Round 1

Reviewer 1 Report

In the manuscript presented here, the authors examine the extent and quality of bio-risk management measures in various life science laboratories at an Indonesian university. Various laboratory facilities were examined in a standardized survey, and further information from secondary sources was evaluated. As a result, the study describes a degree of implementation of bio-risk management measures that is still insufficient in parts in several of the investigated laboratories. The paper provides important information for further development to improve a standardized laboratory risk management policy.

The study is well planned and especially the methodological aspects are described in detail. The evaluation is also comprehensive and reported in full depth. The substantive results, especially the still large potential for improvement in the area of bio risk management in almost all laboratories studied, are not surprising. More such surveys are a compelling prerequisite for supporting, in particular, political decision-making processes to improve biorisk management around the world. For this very reason, the information contained in the manuscript can also be used as a good template for similar investigations by other research groups. 

Possible imitations of the study are appropriately presented by the authors themselves. However, they should not prevent publication at this stage.

Therefore, I have only a minor comment for further improvement of the presented paper:

The presentation of the results on the individual categories of bio risk management regulations studied in the individual laboratories is quite extensive. Possibly, the corresponding numerous tables could also be provided in an appendix.

Author Response

RESPONSE TO REVIEWER 1 Comments and Suggestions for Authors In the manuscript presented here, the authors examine the extent and quality of bio-risk management measures in various life science laboratories at an Indonesian university. Various laboratory facilities were examined in a standardized survey, and further information from secondary sources was evaluated. As a result, the study describes a degree of implementation of bio-risk management measures that is still insufficient in parts in several of the investigated laboratories. The paper provides important information for further development to improve a standardized laboratory risk management policy. The study is well planned and especially the methodological aspects are described in detail. The evaluation is also comprehensive and reported in full depth. The substantive results, especially the still large potential for improvement in the area of bio risk management in almost all laboratories studied, are not surprising. More such surveys are a compelling prerequisite for supporting, in particular, political decision-making processes to improve bio-risk management around the world. For this very reason, the information contained in the manuscript can also be used as a good template for similar investigations by other research groups. Possible imitations of the study are appropriately presented by the authors themselves. However, they should not prevent publication at this stage. Therefore, I have only a minor comment for further improvement of the presented paper: The presentation of the results on the individual categories of bio risk management regulations studied in the individual laboratories is quite extensive. Possibly, the corresponding numerous tables could also be provided in an appendix. Responses: Thank you for the great comments. It really helps us to improve or paper. Since the clauses of the ISO 35001:2019 we put on the table per each clause. So the readers are easy to follow. Thank you for suggestion

Reviewer 2 Report

- The grammar used in the introduction makes it very difficult to read and understand the points that the author is making. A rewrite of the introduction is absolutely needed

- I find the topic and the observation study interesting and worthwhile to publish

Author Response

RESPONSE TO REVIEWER 2

Comments and Suggestions for Authors

  • The grammar used in the introduction makes it very difficult to read and understand the points that the author is making. A rewrite of the introduction is absolutely needed
  • I find the topic and the observation study interesting and worthwhile to publish

Response:

Thank you for great comments and suggestion Sir/Madame. We have updated the introduction based on your suggestion.

Reviewer 3 Report

  1. Line 50: Suggest to explain the scoring as
    1. Yes – meeting all requirements with documented evidence of full implementation
    2. Partial - not meeting all requirement fully
    3. NO - no evidence of implementation
  2. Line 154
    1. Suggest formular as Rate of Implementation (%) =  Score Obtained/Total possible scores x 100
  3. Results -  The total scores are very small for each of the 7 areas. I suggest coming up with composite tables rather that the many tables with scores of 2 as 100% and 1 as 50%. This does not provide any additional meaning. Tables can be collapsed into composite tables to provide more meaningful results interpretation e.g. table 10 is better to read and provides more meaning
  4. Table 10 can be enhanced by indicating scores per section and % to give context to the score. A graph on % can then be drawn and a cut-off line put to show areas performing below 50%. These are the areas that can then be reviewed in the discussion area

Author Response

RESPONSE TO REVIEWER 3

Comments and Suggestions for Authors

  1. Line 50: Suggest to explain the scoring as
    1. Yes – meeting all requirements with documented evidence of full implementation
    2. Partial - not meeting all requirement fully
    3. NO - no evidence of implementation

Response:

Thank you for suggestions. We have already updated in line 149-150.

Line 154

Suggest formular as Rate of Implementation (%) =  Score Obtained/Total possible scores x 100

Response:

Thank you for suggestions. We have already updated based on your suggestion.

Results -  The total scores are very small for each of the 7 areas. I suggest coming up with composite tables rather that the many tables with scores of 2 as 100% and 1 as 50%. This does not provide any additional meaning. Tables can be collapsed into composite tables to provide more meaningful results interpretation e.g. table 10 is better to read and provides more meaning

Response:

Thank you for suggestion. We have updated and made it according to table 3. All sub-clauses were put on in one table. Any items that have not implemented was highlighted with red areas.

Table 10 can be enhanced by indicating scores per section and % to give context to the score. A graph on % can then be drawn and a cut-off line put to show areas performing below 50%. These are the areas that can then be reviewed in the discussion area

Response: Thank you for suggestion. We updated it according to table 4.

Reviewer 4 Report

Dear Authors

This paper is of great importance to the laboratories in developing countries but needs major revision as follows:

Overall: Exactly what are you trying to convey to the reader? This is not clear. However, what you probably should convey is that in your University you did a survey on what elements of Biorisk Management system (35001) has been implemented and to what extent, you have used checklist to do this. The paper should be written in a manner that the readers can use this same method in their own institutions to analyze the gaps and implement a robust BRM system. Please use the accepted term BRM and not BSM

Introduction: not enough references to emphasize your point of view and poor development of the idea of your project, which is the purpose of introduction. Some references are incorrect, no reference to the WHO LBM 4th edition which is very important to this paper. There are several terms in the methods which have not been discussed in the introduction like PDCA, elements of the ISO-35001. These appear suddenly in the materials and methods

Materials and methods: you say you have used interviews and document review, but these are not described. The detailed tables give no useful information, what was in the checklist to actually assess the situation, what question was asked in the checklist? The tables are just listing the elements of the ISO 35001, without explaining what information you actually gathered. Lots of repeat and unnecessary details. Who did you send the checklist to? The heads of the labs or the staff working there?

Results;

Based on the materials and methods it is not possible to understand how you derived at the results, there is no relationship. How did you score the results what type of answer got what score? Did you only use the checklist based on the declaration of the participants? Did you do any document review and verification? A lot of data that is not relevant to the ideas you are projecting in the project

Discussion;

The discussion is not supported by the data, what you need to give is very clear idea of what the survey showed, where the gaps were, how did you determine where the gaps were and what you will be doing about it.

The statements in the discussion cannot be derived from the results you have given. For this paper to be useful, it should be written in a way that another institution can actually follow your method to identify gaps and implement a better BRM in their institution.

Author Response

RESPONSE TO REVIEWER 4

Comments and Suggestions for Authors

Dear Authors

This paper is of great importance to the laboratories in developing countries but needs major revision as follows:

Overall: Exactly what are you trying to convey to the reader? This is not clear. However, what you probably should convey is that in your University you did a survey on what elements of Biorisk Management system (35001) has been implemented and to what extent, you have used checklist to do this. The paper should be written in a manner that the readers can use this same method in their own institutions to analyze the gaps and implement a robust BRM system. Please use the accepted term BRM and not BSM

Response: Thank you for the great feedback. We have tried to updated it according to your notes in order to improve the quality of our paper. The term of BSM have changed to BRM.

Introduction: not enough references to emphasize your point of view and poor development of the idea of your project, which is the purpose of introduction. Some references are incorrect, no reference to the WHO LBM 4th edition which is very important to this paper. There are several terms in the methods which have not been discussed in the introduction like PDCA, elements of the ISO-35001. These appear suddenly in the materials and methods

Response:

Thank you for the comments. We rewrote the part of introduction and considered additional references according to your advices. Such as reference to WHO LBM 4th. “According to 4th Laboratory Biosafety Manual proposed by World Health Organization, biosafety and biosecurity activities are crucial to be managed properly in order to preventing the laboratory employees and surrounding community against unintentional exposures or spread of pathogenic biological agents. Hence, risk assessment framework needs to be implemented as a control measures to minimize the possibility and consequences of any potential exposure to biological agents in laboratory. The framework started from gather the information to review risks and risk control measures [12].”.

PDCA-->. This standard also emphasized on the plan-do-check-act (PDCA) system principle, which highlighted the sequence process of identification, assessment, control and monitor the available risks and hazards associated with biological substances in laboratory within ten clauses, from scope to continuous improvement.Other updates could be found on new manuscript.

Materials and methods: you say you have used interviews and document review, but these are not described. The detailed tables give no useful information, what was in the checklist to actually assess the situation, what question was asked in the checklist? The tables are just listing the elements of the ISO 35001, without explaining what information you actually gathered. Lots of repeat and unnecessary details. Who did you send the checklist to? The heads of the labs or the staff working there?

Response:

Thank you, sir/madam. We apologized there were some mistakes. The study focusses on the checklist assessment and literature review. We also could not explain in detail each question since there was 202 items. Basically, these items were the part of question in each clause as we sent in additional document. We also added the explanation the information of each clause in Table 2.  We updated “This study is a semiquantitative descriptive study that analyses the bio-risk implementation evaluation in laboratories of University Y. Primary data were collected through direct observation, BRM instrument checklist, and interview to the head of laboratories. The survey was performed by principal investigator and a Professor whom experts in Occupational and Health Safety and certified in BRM. Moreover, the secondary data were obtained through literature review. The study was performed during the period of May – June 2020. The theoretical framework of this present study is illustrated in Figure 1”

Results;

Based on the materials and methods it is not possible to understand how you derived at the results, there is no relationship. How did you score the results what type of answer got what score? Did you only use the checklist based on the declaration of the participants? Did you do any document review and verification? A lot of data that is not relevant to the ideas you are projecting in the project

Response:

Thank you for the concern, since we updated a little bit in the method section, the results derived from score of checklist only, hence we refers to Data processing, which was done by calculating each yes (meeting all requirements with documented evidence of full implementation), partial (not meeting all requirement fully), and No (No evidence of implementation) responses with a score of 1, 0.5 and 0 respectively. We also updated the result to make it clear.

Discussion;

The discussion is not supported by the data, what you need to give is very clear idea of what the survey showed, where the gaps were, how did you determine where the gaps were and what you will be doing about it.

The statements in the discussion cannot be derived from the results you have given. For this paper to be useful, it should be written in a way that another institution can actually follow your method to identify gaps and implement a better BRM in their institution. 

Response:

Thank you for the concern. Basically, the gaps were clearly mentioned in results section. We also discussed in any order based on the clause what is the gap and supported references such as

On the results for each clause above, a gap analysis was carried out in order to identify the aspects that were not met yet so that they could be prioritized immediately for continual improvement. Gap analysis identified the items that had been not implemented from 202 items of ISO 35001:2019. As far as the gap analysis score was concerned, 3 clauses with the highest gaps were leadership, performance evaluation, and support (Table 4). Furthermore, the highest gaps of leadership were identified in policies and organizational roles, responsibilities, and authorities. Meanwhile, the lowest gaps were in the clauses of the organizational context, planning, operations, and improvements.

In the leadership clause, policy is the major contributor in effective implementation of BRM. Our finding shows that policies are still being made in general, namely under the policies related to the Occupational Health and Safety of the laboratories. This means that no specific policy is developed for bio-risk. In the policy development, the first step that can be taken is to identify the existing hazards and risks in the laboratory as the content of the policy must be in accordance with the nature and scale of risks associated with the facilities and activities in the laboratory. Actions taken by the top management demonstrate a commitment to policies on laboratory bio-risk management. The policies that cover all areas of work should be assessed on the basis of risk and carried out prior to giving approval to commence any kind of tasks on hazardous biological substances. These policies include protection of all personnel; risk reduction of biological agents; required regulations, information and communication; and continual improvement [29].

Additional information also added to support and identify what is the gap and how to improve

In the planning clause, the gap was found in the sub-clause of bio-risk management objective. In this issue, the organization had not established a document of bio-risk management objectives at relevant functions and levels. According to the ISO 35001: 2019 standard, the objective should meet the requirements such as the consistency of objectives to the implementation, how the document is reviewed, communicated, and updated. Hence, laboratories are also required to prepare risk assessment documents for each activity carried out as planning systems.

Round 2

Reviewer 4 Report

Dear Authors

Overall: This version is better but needs some revisions

Introduction: This version is better

Materials and methods:

Figure 1 is unnecessary, and Table 1 does not add any information that is already described in the text. The results were gathered based on the survey where the answers could be Yes, partial, or No, was there another method of gathering data?

Results:

Please see my comments in the document  

Discussion:

Discussion is very long and a lot of information that is not useful. Reduce to the following:

  1. Overall score
  2. Discussion on the clauses/elements that were good and bad with some possible explanations
  3. What can you do about the clauses/elements that are not good and how you can improve the ones that are good but can still improve?

Author Response

RESPONSE TO REVIEWER 4 ROUND 2

Overall: This version is better but needs some revisions

Response: Thank you for you feedback.

Introduction: This version is better

 Response: Thank you for you feedback.

Materials and methods:

Figure 1 is unnecessary, and Table 1 does not add any information that is already described in the text. The results were gathered based on the survey where the answers could be Yes, partial, or No, was there another method of gathering data?

Response: Thank you for your review. Figure 1 has been deleted. We provide Table 1 to depict this sentence “The inclusion criteria used in this study were the laboratories that involved the use of biological materials in its activities, resulting in 11 laboratories included as the subjects in this study (Table 1). The laboratories that use biological materials are located in several faculties: Faculty of Mathematics and Natural Science, Faculty of Engineering, and in the Health Science cluster”. The gathering data was based on the result of BRM instrument checklist that has been assessed regarding the implementation of each clause. Then, Data processing was done by calculating each yes (meeting all requirements with documented evidence of full implementation), partial (not meeting all requirement fully), and No (No evidence of implementation) responses with a score of 1, 0.5 and 0 respectively. The implementation can be observed through the evidence such documents, documented programs and so on. The total score was calculated using the formula mentioned on the manuscript. Table 2 was deleted, and we described in a text.

Results:

Please see my comments in the document  

Response: We revised according to your notes.  Lab I, J, K (Wet Lab 1, 3, 4) in Health Science Cluster. Since these laboratories are under same management and characteristics, it was grouped into one category. Grey areas were the three highest gaps found in each laboratory according to each element of ISO 35001:2019; red area was the lab with bad implementation of ISO 35001:2019.

Discussion:

Discussion is very long and a lot of information that is not useful. Reduce to the following:

  1. Overall score
  2. Discussion on the clauses/elements that were good and bad with some possible explanations
  3. What can you do about the clauses/elements that are not good and how you can improve the ones that are good but can still improve?

Response: Thank you for the comment and advice. We have already  updated it.

It is clear that laboratories in educational setting have a potential to negatively impact users, or even communities, due to potential incidents caused by their use of biological subtances.  The present study aims to examine the implementation of bio-risk management system in accordance to the recent international standard of ISO 35001: 2019 that observed seven main clauses: organizational context, leadership, planning, support, operations, performance evaluation and improvement.

According to the survey, overall, each laboratory needs to enhance the implementation of BRM, particularly in the aspects of leadership, support and performance evaluation. As far as each clause was concerned, the implementation of in the context and organization clause.

We also reduced some sentence as your recommendation.
